# Cell-intrinsic regulation of phagocyte function by interferon lambda during pulmonary viral, bacterial super-infection

**Danielle Antos[1,2], Olivia B. Parks[1], Alexis M. Duray[1,2], Nevil Abraham[2], Joshua J. Michel[1], Saran Kupul[1], Rosemary Westcott[1], John F. Alcorn[1,2]***

**1** Department of Pediatrics, UPMC Children's Hospital of Pittsburgh, Pittsburgh, Pennsylvania, United States of America, **2** Department of Immunology, University of Pittsburgh, Pittsburgh, Pennsylvania, United States of America

* john.alcorn@chp.edu

**Data Availability Statement:** All raw data are attached as excel files as supplementary information. RNA sequencing data is publicly

## Abstract

Influenza infections result in a significant number of severe illnesses annually, many of which are complicated by secondary bacterial super-infection. Primary influenza infection has been shown to increase susceptibility to secondary methicillin-resistant *Staphylococcus aureus* (MRSA) infection by altering the host immune response, leading to significant immunopathology. Type III interferons (IFNs), or IFNλs, have gained traction as potential antiviral therapeutics due to their restriction of viral replication without damaging inflammation. The role of IFNλ in regulating epithelial biology in super-infection has recently been established; however, the impact of IFNλ on immune cells is less defined. In this study, we infected wild-type and IFNLR1⁻/⁻ mice with influenza A/PR/8/34 followed by *S. aureus* USA300. We demonstrated that global IFNLR1⁻/⁻ mice have enhanced bacterial clearance through increased uptake by phagocytes, which was shown to be cell-intrinsic specifically in myeloid cells in mixed bone marrow chimeras. We also showed that depletion of IFNLR1 on CX₃CR1 expressing myeloid immune cells, but not neutrophils, was sufficient to significantly reduce bacterial burden compared to mice with intact IFNLR1. These findings provide insight into how IFNλ in an influenza-infected lung impedes bacterial clearance during super-infection and show a direct cell intrinsic role for IFNλ signaling on myeloid cells.

## Author summary

Influenza virus infects millions of people each year and is characterized by a mostly self-resolving mild to moderate illness. However, severe disease is observed in a subset of patients and is often characterized by bacterial super-infection, which occurs following influenza virus infection. In this setting, there can be a lethal synergy between the virus and bacteria that results in severe lung injury. In recent years, the field has determined that type III interferon (IFNλ), induced by influenza virus as part of the anti-viral response, may play a detrimental role during super-infection. This has been attributed to IFNλ impacting structural cells of the airways. In this study, we define a novel role for

available on NIH Gene Expression Omnibus (GSE268994).

**Funding:** This grant was supported by NIH NHLBI R01 HL107380 (JFA), NIH NHLBI F31 HL164031 (DA), NIH NHLBI F30 HL159915 (OBP), and the UPMC Children's Hospital of Pittsburgh Research Advisory Committee (DA). The funders had no role in study design, data collection and analysis, decision to publish, or preparation of the manuscript.

**Competing interests:** The authors have declared that no competing interests exist.

IFNλ impacting bacterial clearance during super-infection via directly altering lung phagocytes responsible for bacterial host-defense. We use three distinct mouse models to define this new role of IFNλ in regulating phagocyte function in the lung. This study is important because it defines a new mechanism by which anti-viral IFNλ leads to increased susceptibility to bacterial super-infection. This has implications for potential use of IFNλ as an anti-viral therapeutic and identifies a new biologic function of IFNλ in the lung.

## Introduction

Since their discovery in 2003, type III interferons or interferon (IFN) λs have been studied extensively for their potent antiviral activity [1,2]. While their restriction of viral replication is similar to that of type I IFN, IFNλs do not induce excessive inflammation seen with type I IFNs and induce overlapping but distinct sets of interferon stimulated genes (ISGs) [3–7]. IFNλ administration in both mice and humans has been shown to improve outcomes against multiple respiratory viruses, including influenza virus and, more recently, SARS-CoV-2 [3,4,8–15]. Prophylactic treatment with IFNλ in mice has been shown to prevent infection entirely [16]. Additionally, experimental models of viral infection have shown that IFNλ may represent the first line of defense and is indispensable for promoting antiviral activity, although type I IFNs can compensate in IFNLR1$^{-/-}$ models [3,17–19]. These anti-viral benefits of IFNλ have enhanced its potential as a therapeutic against a number of additional pathogens. Data also increasingly show that IFNλ can effectively act as an adjuvant when given with the influenza vaccine and synergistically if administered with oseltamivir [20,21].

More recently, knowledge regarding the activity of IFNλs has been expanded to include functional roles during bacterial infections. While this field is newer and still very active, roles for IFNλ are more varied against different types of bacteria. Multiple bacteria, including *Staphylococcus aureus*, *Salmonella typhimurium*, and *Listeria monocytogenes*, have been shown to induce IFNλ production, but reports conflict as to whether IFNλ promotes or inhibits bacterial clearance in mice, suggesting that IFNλ may have differential effects depending on pathogen or infection site [22–29]. These dissonant results, when combined with the strong antiviral activity of IFNλs, has led to strong interest in elucidating how IFNλ may contribute to the onset, resolution, or prolongation of viral associated bacterial super-infections.

Bacterial super-infection most often occurs following primary viral infection and replication in the lung, where an inflammatory environment, lung damage, immune cell influx, and other factors promote bacterial outgrowth [30]. Super-infections increase the risk of mortality and long-lasting effects on the lung, shown most prominently during the 1918 and 2009 H1N1 influenza pandemics [31,32]. Mouse models of super-infection have identified dysregulated and over-active immune responses that lead to increased bacterial burden compared to single infection, alterations in antibacterial immune responses, higher lung pathology, and increased mortality [33–36].

Published work has shown multiple facets of IFNλ's negative impact during super-infection. IFNλ has been shown to slow epithelial cell repair after influenza infection, increase bacterial burden in mice during super-infection and single infection, and induce the expression of immunosuppressive molecule, indoleamine 2-3-dioxygenase (IDO) [36–40]. Many studies surrounding IFNλ have historically focused on epithelial cells lining barrier sites, as these were first postulated as specifically expressing the IFNλ receptor [41]. While IFNλ signaling in epithelial cells has been widely studied, the activity of IFNλs on immune cells are an emerging area of research as additional cell types expressing IFNLR1 have been identified [42–45]. Here,

we examined activation of antibacterial immune responses including bacterial uptake and pha-golysosomal localization during super-infection in IFNLR1$^{-/-}$ and wild type (WT) mice to demonstrate direct effects of IFNλ signaling on phagocytes. We then examined specific dele-tion of IFNLR1 both on neutrophils and myeloid cells using mixed bone marrow chimeras and the Cre-LoxP system to identify cell intrinsic effects of IFNλ on specific cell types.

## Results

### Global IFNLR1 knockout mice have decreased bacterial burden during super-infection compared with wild type mice

To determine the impact of IFNλ during super-infection, we first elucidated general differ-ences between C57BL/6 wild type (WT) and global IFNLR1 knockout mice using a 7 day super-infection model, where mice are inoculated with 900 plaque forming units (PFU) of mouse-adapted H1N1 strain A/PR/8/34 on day 0 and with 5 x 10$^7$ colony forming units (CFU) of *Staphylococcus aureus* on day 6 before harvesting 6 or 24 hours after bacterial infection (Fig 1A). Both infection doses have been previously shown to be sub-lethal in WT mice [33,46]. Global IFNLR1$^{-/-}$ mice had significantly reduced bacterial burden versus WT mice at 24 hours post-bacterial infection, with a trend apparent as early as 6 hours post-infection (Figs 1B and S1A). However, several other parameters measuring super-infection severity, including lung leak, weight loss, and immune cell recruitment to the airways, remained unchanged between WT and global IFNLR1$^{-/-}$ mice at both 6 and 24 hours post-bacterial infection (Figs 1C–1E and S1B–1D). Measurement of chemokines in the lung showed several alterations in global IFNLR1$^{-/-}$ mice compared to WT mice both during super-infection and single *S. aureus* infec-tion, including increased TNFα, CXCL11, CXCL12, and MIP3α and decreased IL-1β, MIP1β, and CXCL16 (Fig 1F). Chemokine measurement 6 hours post bacterial infection showed simi-lar trends in some cytokines, but also highlighted the importance of cytokine kinetics within infection, as multiple chemokines were altered only at early time points, including CCL17, CCL22, IL-6, and CXCL5 (S1E Fig). Pathology scoring of lung histology also showed decreased peribronchial inflammation in global IFNLR1$^{-/-}$ mice as compared to WT mice (S2A–2B Fig). These data show that IFNλ signaling modulates bacterial clearance and cytokine production but does not universally impact the inflammatory state in the lung during super-infection. This effect on bacterial clearance suggests a potentially immune cell-intrinsic impact of IFNλ in potentiating super-infection.

### Anti-bacterial type 17 immunity is suppressed by IFNλ signaling

Our laboratory has previously shown that type 17 immune responses, while required for *S. aureus* clearance from the lung, are significantly inhibited during super-infection, in part through type I IFN signaling [33]. Because the IFNλ signaling pathway shares many molecules (ex. Jak/STAT) with type I IFNs, we investigated if type 17 immunity is altered in the presence or absence of IFNλ signaling in a way that may explain enhanced bacterial clearance in these mice. In global IFNLR1$^{-/-}$, IL-17 and IL-22 protein levels were increased in the airways of super-infected mice (Fig 2A). We then examined if the increased bacterial clearance seen in super-infected IFNLR1$^{-/-}$ mice was IL-17-dependent via an antibody blockade of IL-17A. Treatment with anti-IL-17A in super-infected global IFNLR1$^{-/-}$ mice did result in a significant increase in bacterial burden compared to untreated IFNLR1$^{-/-}$ mice (Fig 2B). Consistent with reduced IL-17A, total numbers and frequencies of neutrophils recruited to the lung were sig-nificantly decreased in anti-IL-17A-treated mice, which may explain decreased bacterial

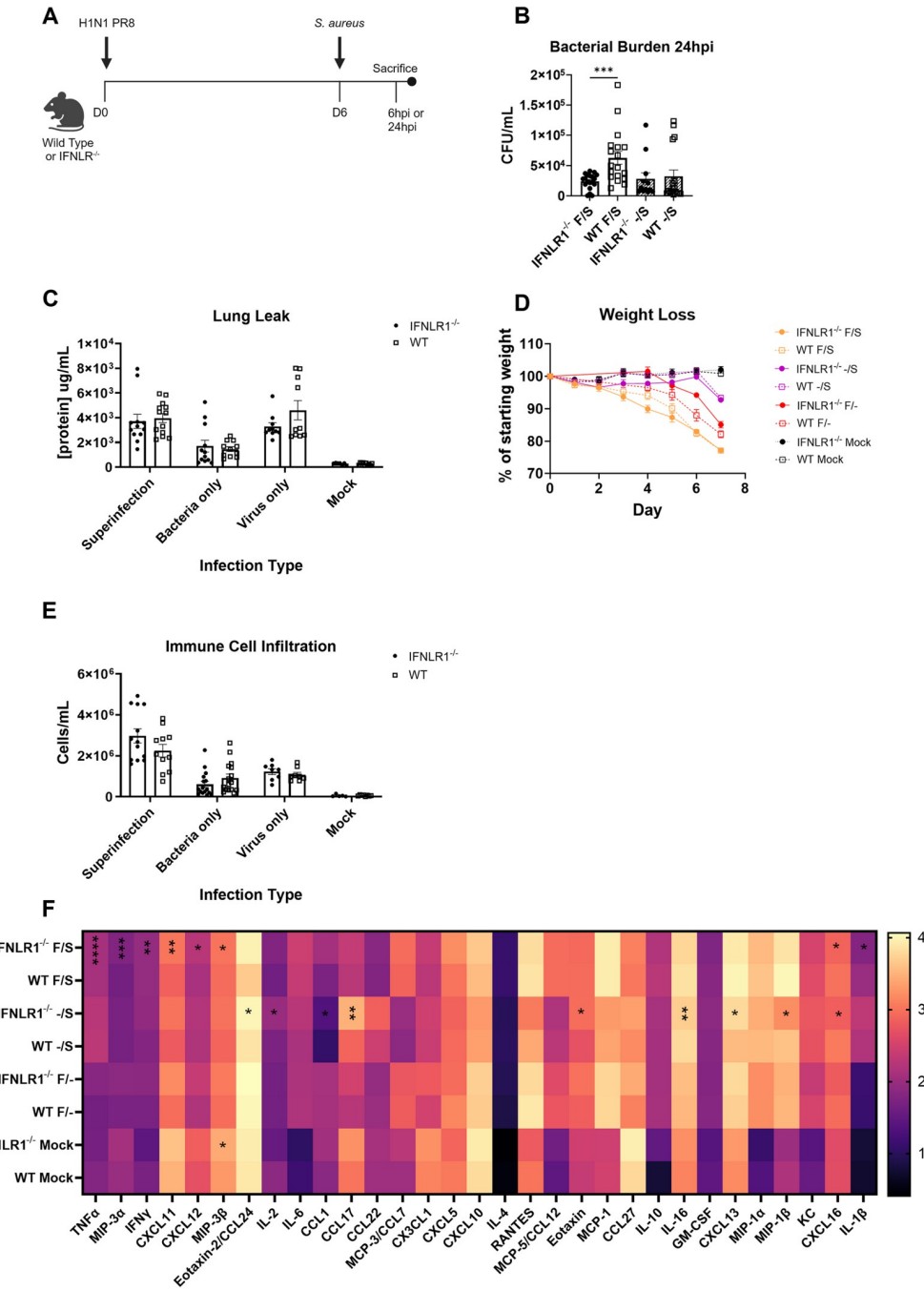

**Fig 1. Global IFNLR1$^{-/-}$ mice have increased bacterial clearance compared with wild type (WT) mice during pulmonary super-infection.** A. Mice were infected with 900 PFU A/PR/8/34 on day 0 and 5 x 10$^7$ colony forming units (CFU) *Staphylococcus aureus* strain USA300 on day 6, followed by harvest on day 7. Figure created on Biorender. com. B. Global IFNLR1$^{-/-}$ mice have reduced bacterial burden compared to WT mice 24 hours post bacterial infection (IFNLR1$^{-/-}$: F/S n = 18, -/S n = 12; WT: F/S n = 17, -/S n = 16). C. Protein levels in the airways of infected mice are not different between IFNLR1$^{-/-}$ and WT mice during super-infection (IFNLR1$^{-/-}$ n = 13, WT n = 13), single bacterial (IFNLR1$^{-/-}$ n = 12, WT = 11) or viral infection (IFNLR1$^{-/-}$ n = 8, WT n = 9), or at basal levels (IFNLR1$^{-/-}$ n = 7, WT n = 10). D. Mouse weight loss was monitored daily during infection time-course (data from at least 3 replicates). E. Immune cell infiltration into the airways was assessed by quantification of cells in bronchoalveolar lavage (BAL) fluid during super-infection (IFNLR1$^{-/-}$ n = 13, WT n = 11), single bacterial (IFNLR1$^{-/-}$ n = 12, WT = 12) or viral infection (IFNLR1$^{-/-}$ n = 8, WT n = 8), or at basal levels (IFNLR1$^{-/-}$ n = 7, WT n = 8). F. Global cytokine levels were detected using Bio-Plex assays in mice on day 7 of the model during super-infection (IFNLR1$^{-/-}$ n = 12, WT n = 12), single infections (Bacteria: IFNLR1$^{-/-}$ n = 5, WT n = 6; Virus: (IFNLR1$^{-/-}$ n = 4, WT n = 4), and at basal levels (IFNLR1$^{-/-}$

n = 2, WT n = 2). Statistics compare IFNLR1$^{-/-}$ to WT mice within each infection type. p values: *<0.05, **<0.01, ***<0.001, ****<0.0001.

clearance (Fig 2C). This indicates that increased bacterial clearance seen during super-infection in global IFNLR1$^{-/-}$ mice is associated with enhanced IL-17 production.

Next, to identify how changes in IL-17 and IL-22 induction may be occurring *in vivo*, concentrations of molecules known to influence type 17 immune cells were examined, namely IL-23, IL-1β, and IL-6 [47–49]. Interestingly, global IFNLR1$^{-/-}$ mice had decreased levels of IL-1β compared to controls, and there were no changes in IL-23 or IL-6 levels between super-infected IFNLR1$^{-/-}$ and WT mice (Fig 2D). Although canonical regulators of IL-17 and IL-22 were not altered in global IFNLR1$^{-/-}$ mice, chemokines involved in type 17 cell recruitment and activation were found to be modulated in super-infected IFNLR1$^{-/-}$ mice compared to WT, including CXCL16 and MIP3α (Fig 1F) [50–52]. While the specific molecules driving IL-17 production during super-infection are still unclear, our results suggest that enhanced bacterial clearance in global IFNLR1$^{-/-}$ mice is associated with a more robust type 17 response, which is likely not regulated by one molecule alone.

## IFNλ signaling impacts bacterial uptake by phagocytic cells during super-infection

A common mechanism by which type 17 immunity may increase bacterial host defense is through effects on polymorphonuclear and myeloid phagocytes. Neutrophils are known regulators of type 17 immunity and have been previously shown to have altered function in response to exogenous IFNλ, but the requirement of IFNλ signaling in suppression of phagocyte function during super-infection has not been demonstrated [39,53,54]. Before undertaking *in vivo* testing, we isolated CD11c+ cells from naïve WT and global IFNLR1$^{-/-}$ mice and confirmed that these cells express *ifnlr1* transcript (S2C Fig). CD11c is expressed on multiple phagocytic populations within the lung, including alveolar macrophages, interstitial macrophages, dendritic cells, and lung monocytes [55].

In order to parse out cell-specific impacts of IFNλ, we used a dsRed expressing *S. aureus* strain to identify whether bacterial uptake is impacted in any immune population in the absence of IFNλ signaling by flow cytometry. Overall, phagocytic cells from IFNLR1$^{-/-}$ mice, including macrophages and monocytes, neutrophils, and dendritic cells, had a significantly higher frequency and total number of dsRed+ cells at both 6 and 24 hours post bacterial infection without any alteration in recruited numbers into the lung (Figs 3A and S3A–3C). There were no changes in the overall frequencies or cell counts of these populations between WT and IFNLR1$^{-/-}$ mice (S3D Fig). These changes in bacterial uptake were consistent with a trend towards increased phagolysomal localization when bacteria were labeled with pH-dependent dye pHrodo red rather than dsRed (S4 Fig). To ensure that this observed uptake was measuring bacteria inside of immune cells and not increased stickiness of bacteria to the outside of immune cells, we performed ImageStream analysis and confirmed that bacteria was internalized into phagocytes (Fig 3B).

We next performed mixed bone marrow chimeras with congenically marked WT and global IFNLR1$^{-/-}$ mice to determine if the impacts of IFNλ signaling on immune cells were direct or indirect in the same host background. This model, where epithelial cell expression of IFNLR1 is uniform, allows for direct comparison of functional activity between IFNLR1$^{-/-}$ and WT immune cells. WT or IFNLR1$^{-/-}$ mice were sub-lethally irradiated and reconstituted with a total of 1 x 10$^7$ bone marrow cells at a 1:1 ratio of WT and IFNLR$^{-/-}$ cells. Mice were super-

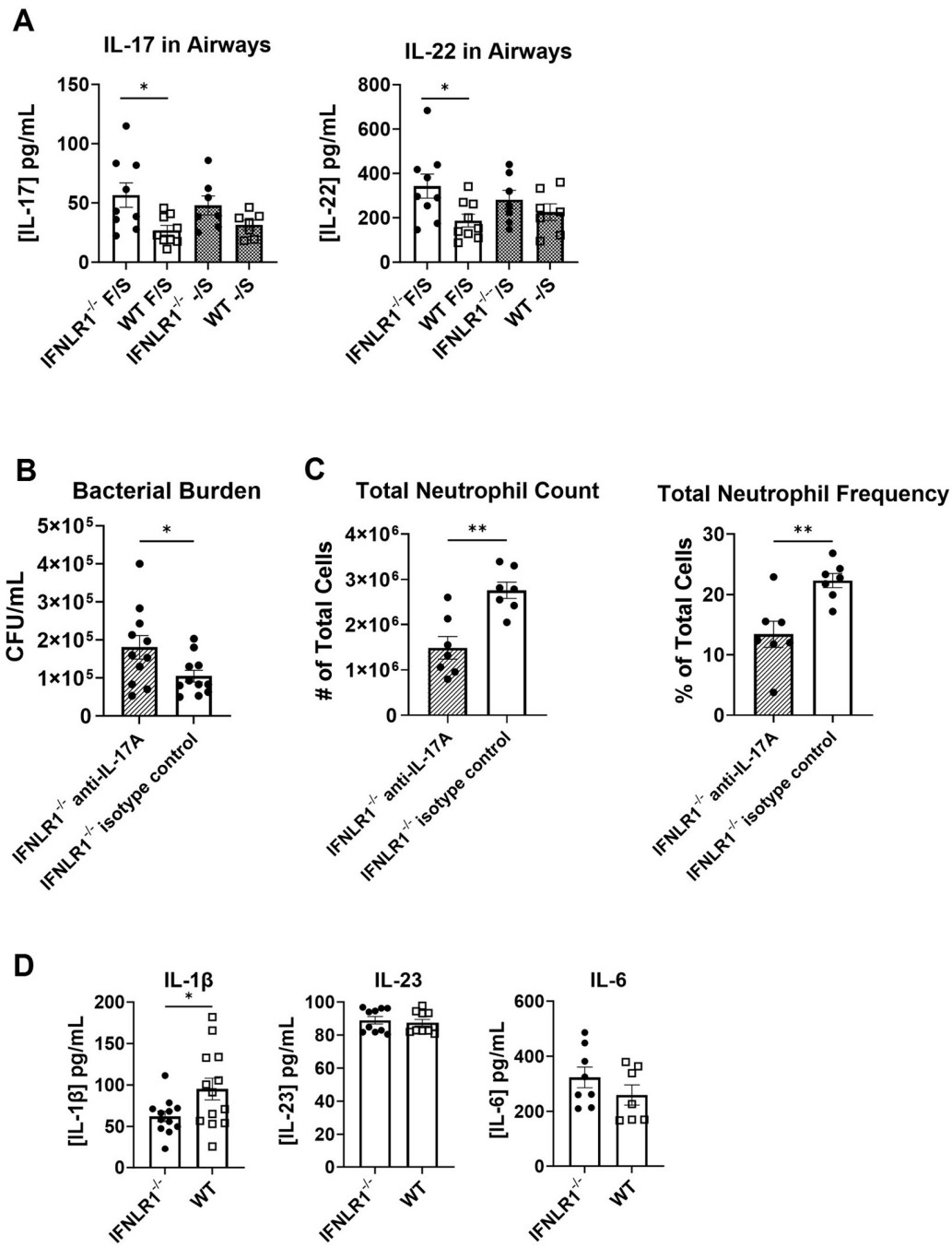

**Fig 2. IFNλ modulates chemokine production to influence antibacterial type 17 immunity in the lung.** A. IL-17 and IL-22 were measured in the BAL fluid by ELISA, and both were increased in super-infected IFNLR1[-/-] mice (F/S: IFNLR1[-/-] n = 9, WT n = 9; -/S: IFNLR1[-/-] n = 7, WT n = 7; data from 3 replicates). B. Antibody blockade of IL-17A increased bacterial burden in super-infected IFNLR1[-/-] mice (Ad-IL-17 n = 11, Ad-eGFP n = 11). C. Neutrophil recruitment to the lung was measured and quantified by flow cytometry in total cell counts (left) and frequency (right; representative data from 2 replicates, n = 7). D. Levels of IL-1β (left; IFNLR1[-/-] n = 12, WT n = 13), IL-23 (middle; IFNLR1[-/-] n = 10, WT n = 10), and IL-6 (right; IFNLR1[-/-] n = 8, WT n = 7) were assessed in super-infected mice by Bio-plex or ELISA (data from 3 replicates). p values: *$<$0.05, **$<$0.01.

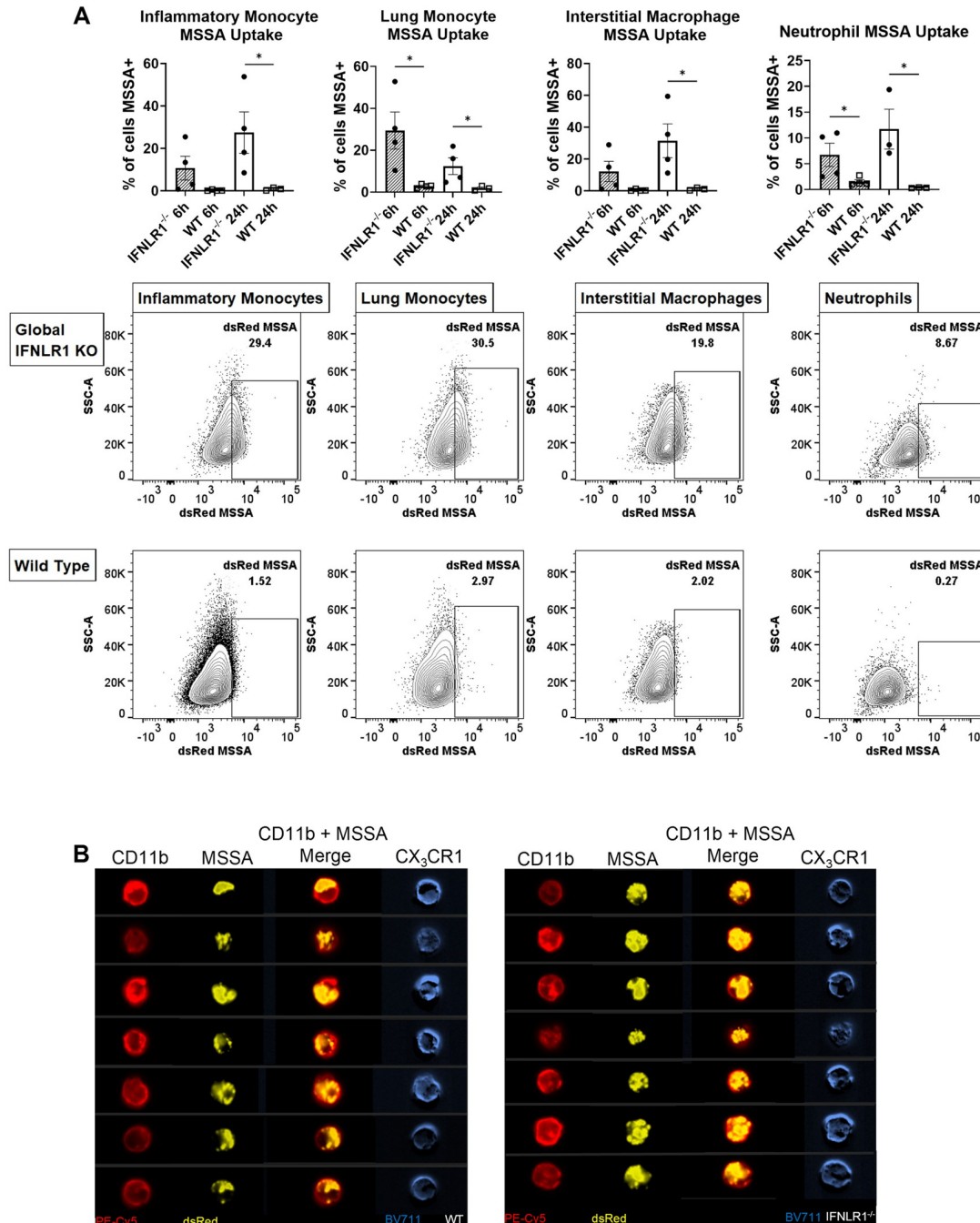

**Fig 3. Disruption of IFNλ signaling in phagocytes directly increases their ability to phagocytose bacteria.** A. Global IFNLR1$^{-/-}$ and WT mice were super-infected as previously stated and fluorescently labeled MSSA strain RN6390 (dsRed expressing) were used to measure and quantify bacteria taken up by phagocytes. IFNLR1$^{-/-}$ mice had increased uptake both 6 hours post bacterial infection (hashed bars) and 24 hours post bacterial infection (open bars), the latter shown in flow plots below (data from 3 replicates). Populations of interest included inflammatory monocytes (CD45$^+$, Siglec-F$^-$, Ly6C$^{++}$, CD11b$^+$, MHC-II$^{+/-}$), lung monocytes (CD45$^+$, CD11b$^+$, Siglec-F$^-$, Ly6C$^{+/-}$, CX$_3$CR1$^+$), interstitial macrophages (CD45$^+$, CD11b$^+$, Siglec-F$^-$, MHC-II$^+$, CX$_3$CR1$^+$), and neutrophils (CD45$^+$, CD11b$^+$, Ly6G$^+$). B. Amnis ImageStream was used to visualize increased phagocytosis in IFNLR1$^{-/-}$ immune cells. IFNLR1$^{-/-}$ cells (left; n = 8) have increased dsRed *S. aureus* staining (yellow) within the cell membrane compared to WT cells (right; n = 7). Cells that stain positively for *S. aureus* also express CX$_3$CR1 highly. p values: *<0.05.

infected as described 9 weeks after irradiation and reconstitution (S5A Fig). There were no differences in weight loss, bacterial burden, immune cell recruitment, or lung leak in the lungs between WT and IFNLR1[-/-] hosts, and reconstitution after nine weeks was similar between IFNLR1[-/-] and WT cells (Figs 4A–4B and S5B). Interestingly, IFNLR1[-/-] myeloid cells, specifically inflammatory monocytes, lung monocytes, and interstitial macrophages, had significantly increased bacterial uptake both by the frequency of dsRed+ cells and the median fluorescence intensity (MFI) without significant changes in the cell population sizes (Figs 4C–4E and S5C). This was seen predominantly in the WT background. Surprisingly, neutrophils were not significantly impacted in this model, as no differences were seen between IFNLR1[-/-] and WT neutrophils in either host (Fig 4C–4E). Together, these data suggest a cell intrinsic effect of IFNλ on several populations of phagocytes to reduce antibacterial activity during super-infection. This conclusion is directly supported by mixed bone marrow chimera, where IFNLR1[-/-] myeloid phagocytes outperform WT phagocytes in a WT mouse host.

## Specific deletion of IFNLR1 on neutrophils does not alter bacterial clearance during super-infection

After identifying both that IFNλ inhibited IL-17 production and likely acted directly on immune cells to decrease phagocytosis, we next sought to elucidate the role of neutrophils more specifically. Data from global IFNLR1[-/-] mice showed increased frequencies of dsRed + cells compared to WT neutrophils, which was inconsistent with data from mixed bone marrow chimeras, where IFNLR1[-/-] and WT neutrophils had no uptake difference in either WT or IFNLR1[-/-] hosts. Because neutrophils are known effectors of the type 17 response and were significantly decreased in anti-IL-17A-treated mice, understanding the impacts of IFNλ specifically on this cell type may explain downstream effects seen during super-infection. IFNLR1[fl/fl] mice generated at the University of Pittsburgh were crossed to S100A8 Cre+ mice, resulting in IFNLR1 depletion that is largely specific to neutrophils and other granulocytes, with low activity seen in populations of monocytes and macrophages [56,57]. Surprisingly, S100A8-Cre-IFNLR1[fl/fl] mice did not have any significant change in bacterial burden or neutrophil-specific bacterial uptake or recruitment to the lungs compared to Cre- littermate controls (Fig 5A–5D). Additionally, S100A8-Cre-IFNLR1[fl/fl] mice did not show any difference in survival after lethal super-infection compared to Cre- littermate controls or in viral burden and morbidity after single viral or bacterial infection (Figs 5E and S6). These results, combined with those from mixed bone marrow chimeric mice, indicate that IFNλ is not acting intrinsically on neutrophils to impede bacterial clearance during super-infection.

## Disruption of IFNλ signaling specifically in lung myeloid cells is sufficient to improve bacterial clearance during super-infection

Because deletion of IFNLR1 specifically on neutrophils did not impact bacterial phagocytosis and mixed bone marrow chimeras suggested a direct effect for IFNλ on myeloid cells, we next deleted IFNLR1 specifically on these cells. Lung myeloid cells from global IFNLR1[-/-] mice also had enhanced bacterial phagocytosis during super-infection compared to WT mice and have been shown to respond to IFNλ signaling, and these cells produce a large number of cytokines and chemokines, including those that induce type 17 immune responses [58–60]. We employed inducible CX$_3$CR1-specific conditional IFNLR1 knockout mice to determine if IFNλ acts directly on these cells during super-infection. CX$_3$CR1 Cre expression was restricted to myeloid cells, including inflammatory monocytes, lung monocytes, interstitial macrophages, and CD11b+ dendritic cells, but not neutrophils, and *ifnlr1* transcript levels were moderately reduced in tamoxifen-treated CX$_3$CR1 Cre+ compared to controls (S7A–7B Fig) [61].

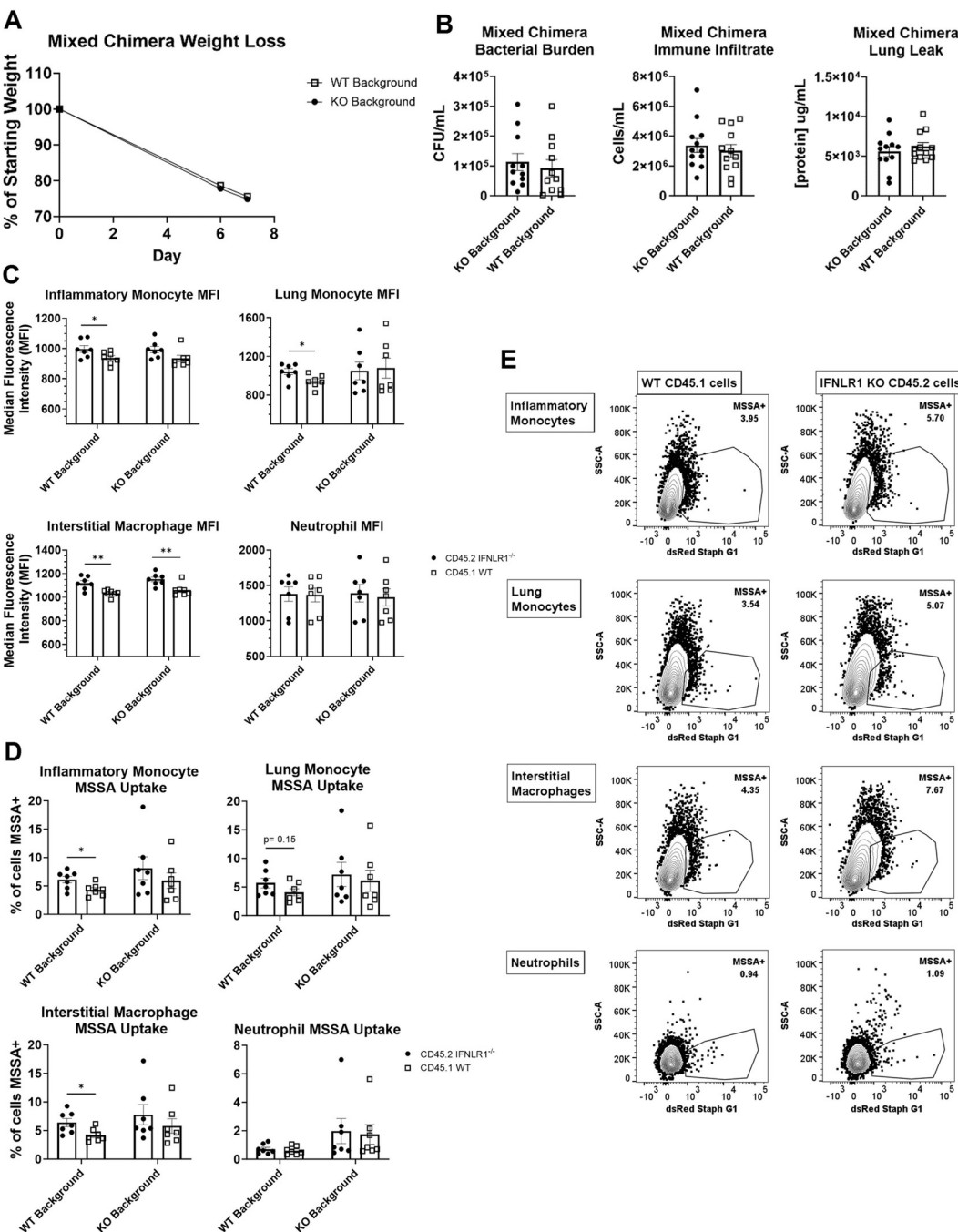

**Fig 4. IFNλ acts directly on immune cells to impede bacterial phagocytosis.** A. Weight loss is equal between WT host mice and IFNLR1$^{-/-}$ host mice. B. Mixed bone marrow chimeras have comparable bacterial burden (left), immune cell infiltration (middle), and lung leak (right) regardless of host genotype (host: KO n = 12, WT n = 12). C. In a WT host, IFNLR1$^{-/-}$ immune cells (inflammatory monocytes, lung monocytes, and interstitial macrophages) take up more bacteria, visualized by MFI, than WT immune cells. Uptake in the IFNLR1$^{-/-}$ host is not significantly altered (host: KO n = 8, WT n = 8). D. In WT hosts, IFNLR1$^{-/-}$ immune cells have increased frequencies of dsRed+ cells compared to WT immune cells (host: KO n = 8, WT n = 8). These trends are consistent but not significant in KO hosts. E. Visualization of increased dsRed+ populations in IFNLR1$^{-/-}$ immune cells compared to WT immune cells within the same WT host. p values: *<0.05, **<0.01.

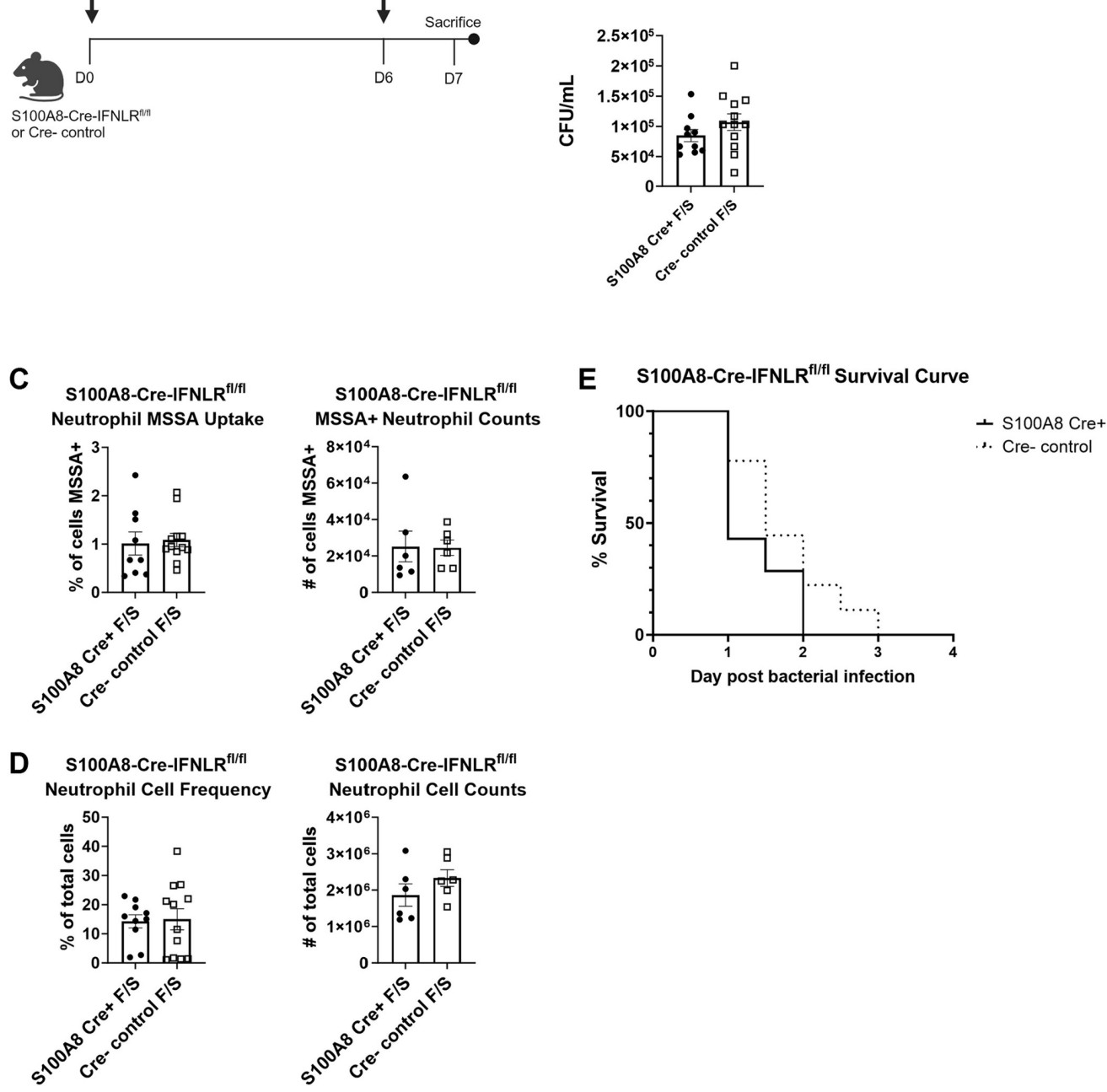

**Fig 5. IFNλ does not directly act on neutrophils to decrease bacterial clearance during super-infection.** A. S100A8-Cre-IFNLR1$^{fl/fl}$ mice constitutively express Cre, and thus were super-infected as previously stated with harvest on day 7. Figure created on Biorender.com. B. S100A8-Cre-IFNLR1$^{fl/fl}$ (cKO) mice do not have significantly altered bacterial burden compared to Cre- littermate controls (cKO n = 10, Cre- n = 12). C. Neutrophil uptake of dsRed-labeled bacteria was not altered by frequency (left; cKO n = 10, Cre- n = 12) or total number (right; cKO n = 6, Cre- n = 6) of dsRed+ cells in S100A8-Cre-IFNLR1$^{fl/fl}$ versus Cre- controls. D. Neutrophil recruitment to the lung, measured by flow cytometry, was not altered by frequency (left; cKO n = 10, Cre- n = 12) or total number (right; cKO n = 6, Cre- n = 6) in S100A8-Cre-IFNLR1$^{fl/fl}$ versus Cre- controls. E. Mortality curve showed no difference in S100A8-Cre-IFNLR1$^{fl/fl}$ or Cre- control survival during lethal super-infection (cKO n = 7, Cre- n = 9).

Experimental Cre+ mice and Cre- littermate controls were injected with Tamoxifen 14 and 7 days before influenza infection, and on day 0 and 5 of the super-infection model (Fig 6A). CX$_3$CR1 conditional knockout mice have reduced bacterial burden compared to Cre-

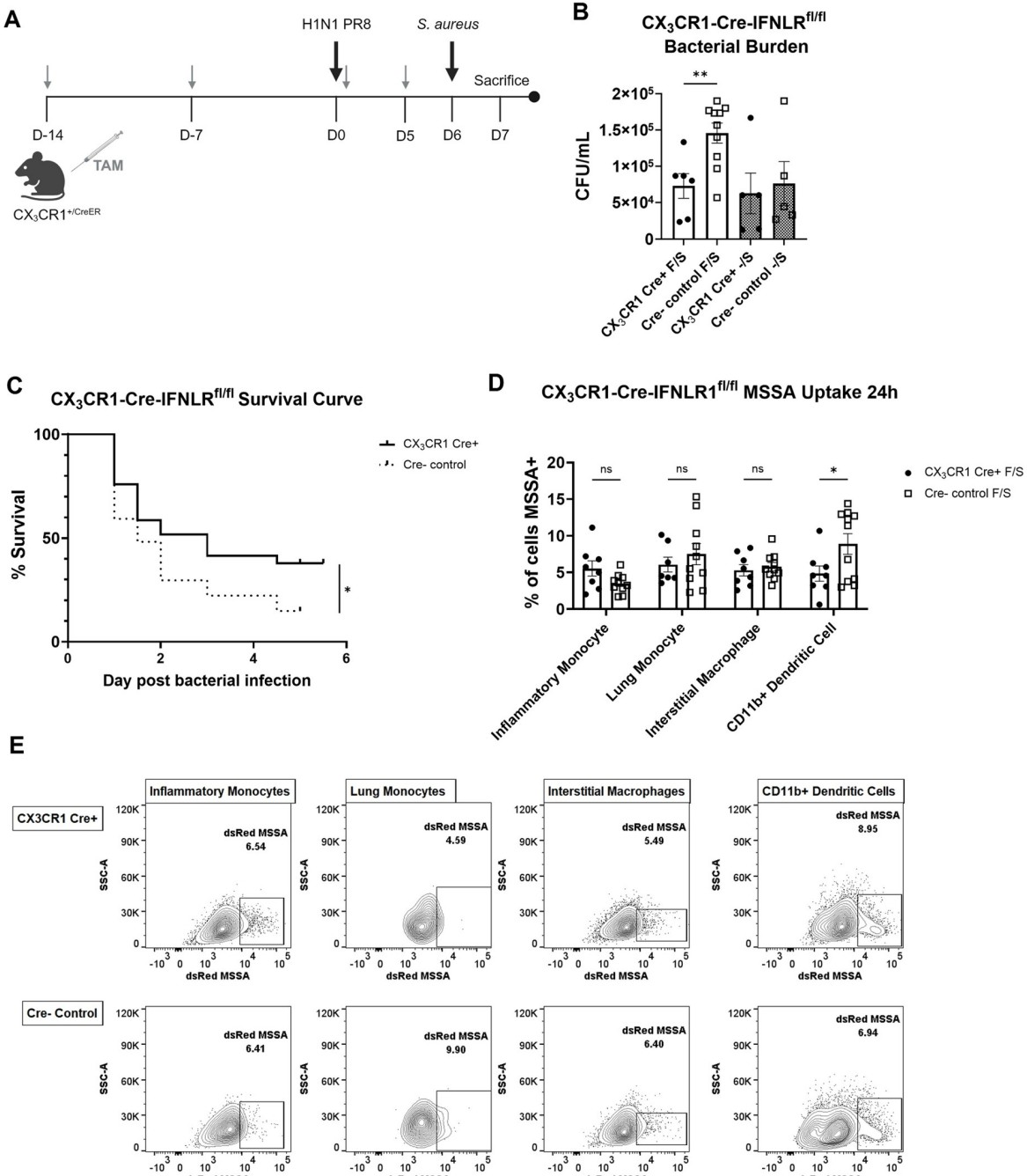

**Fig 6. Depletion of IFNLR1 specifically on CX₃CR1 cells is sufficient to increase bacterial clearance during super-infection.** A. CX₃CR1-Cre-IFNLR1$^{fl/fl}$ mice or Cre- littermate controls were injected with 2mg of Tamoxifen two weeks and one week before influenza infection on day 0. Super-infection proceeded as previously, with additional injections given on days 0 and 5 of the model. Figure created on Biorender.com. B. CX₃CR1-Cre-IFNLR1$^{fl/fl}$ (cKO) mice had significantly reduced bacterial burden during super-infection (open bars; cKO n = 6, Cre- n = 10, data from 3 replicates) compared to littermate controls, but there are no differences during single bacterial infection (hashed bars; cKO n = 5, Cre- n = 5, data from 2 replicates). C. CX₃CR1 cKO mice have greater and longer survival after lethal super-infection compared to littermate controls (p = 0.05; cKO n = 29, Cre- n = 27). D-E. Bacterial uptake by immune cells was largely unchanged in super-infected CX₃CR1 cKO mice and littermate controls 24h post bacterial infection (D, cKO n = 8, Cre- n = 11), also shown by representative flow plots below (E). p values: *<0.05, **<0.01.

littermate controls, phenocopying global IFNLR1$^{-/-}$ mice (Fig 6B). Interestingly, CX$_3$CR1 conditional knockout mice had significantly reduced levels of influenza *m* protein on day 7 of a single viral infection as compared to Cre- controls while still losing similar percentages of body weight and producing similar levels of type I IFNs in the lungs (S7C–7E Fig). Disruption of IFNλ signaling specifically in CX$_3$CR1+ cells protected against mortality during lethal super-infection as compared to Cre- littermate controls (Fig 6C). Levels of several epithelial cell markers indicating tissue health and barrier integrity including surfactant protein C (*sftpc*), club cell secreting protein (*scgb1a1*), and tight junction protein 1 (*tjp1*) were unchanged between CX$_3$CR1 conditional knockout mice and Cre- littermate controls (S7F Fig).

Again, we used dsRed expressing *S. aureus* to quantify bacterial uptake by various phagocytic populations. Unlike global IFNLR1$^{-/-}$ mice, there were no differences in dsRed+ cells between the conditional knockout and Cre- littermate control mice (Fig 6D–6E). To identify whether changes in bacterial uptake were not occurring at earlier time point after *S. aureus* infection, mice were additionally harvested at 6 or 12 hours post bacterial infection. There were no differences between CX$_3$CR1-Cre-IFNLR1$^{fl/fl}$ mice and Cre- littermate controls in total uptake or bacterial burden at either time point (S8 Fig). Together, these data indicate that IFNλ acts directly on lung myeloid cells during super-infection to interfere with bacterial clearance in the lung but does not do so by impeding overall bacterial uptake. Our results suggest that impacts on phagocytosis may require crosstalk between multiple cell types after IFNλ signaling, which is lost in specific CX$_3$CR1 knockout mice.

## IFNLR1$^{-/-}$ phagocytes have increased bacterial uptake into phagolysosomes

Due to the discrepancy between decreased lung bacterial burden and the lack of altered myeloid cell bacterial uptake, we examined an early step in the process of bacterial killing, bacterial phagolysosome localization. To measure changes in elements of bacterial killing and degradation after phagocytosis, *S. aureus* was labeled again with pH-dependent dye, pHrodo red, before infection in CX$_3$CR1-Cre-IFNLR1$^{fl/fl}$ mice. Frequencies of pHrodo red+ cells, which only emit a fluorescent signal in acidic environments, were significantly increased in CX$_3$CR1 + Cre+ phagocytes compared to Cre- controls 6 hours after bacterial infection (Fig 7A). These results show that disruption of IFNλ signaling caused increased inclusion of bacteria into phagolysosomes. This was then confirmed by ImageStream analysis showing co-localization of labeled bacteria with the endosomal marker CD63 (Fig 7B). These results show that IFNλ inhibits an early step in the bacterial degradation pathway.

## IFNλ alters myeloid cell responses during influenza infection

In order to identify other potential pathways by which IFNλ delays bacterial clearance during super-infection, we performed bulk RNA sequencing on myeloid cells from influenza-infected CX$_3$CR1-Cre-IFNLR1$^{fl/fl}$ mice. Before sequencing, target cells were sorted using an eYFP + reporter and control samples consisted of eYFP+ target cells from conditional knockout mice treated with a corn oil control (Fig 8A). This method allowed us to specifically compare CX$_3$CR1+ target cells where IFNLR1 was deleted or not. We identified differentially expressed genes (DEGs) between tamoxifen-treated and -untreated groups and found many genes involved in the regulation of immune responses, including *pacsin1*, *nck2*, and *cd72* (Fig 8B). Within the top 60 interferome-specific DEGs, two variants of the interferon-stimulated gene (ISG) 2'-5' oligoadenylate synthetase 1 (OAS1; *oas1f*, *oas1d*), *ccr6*, *il12a*, and *ctla2b* are amongst genes upregulated in IFNLR1$^{-/-}$ eYFP+ cells (Fig 8C). When all DEGs were analyzed for gene ontology, many pathways upregulated in tamoxifen-treated mice were related to T cell activation and recruitment, immune cell proliferation, and antigen processing and

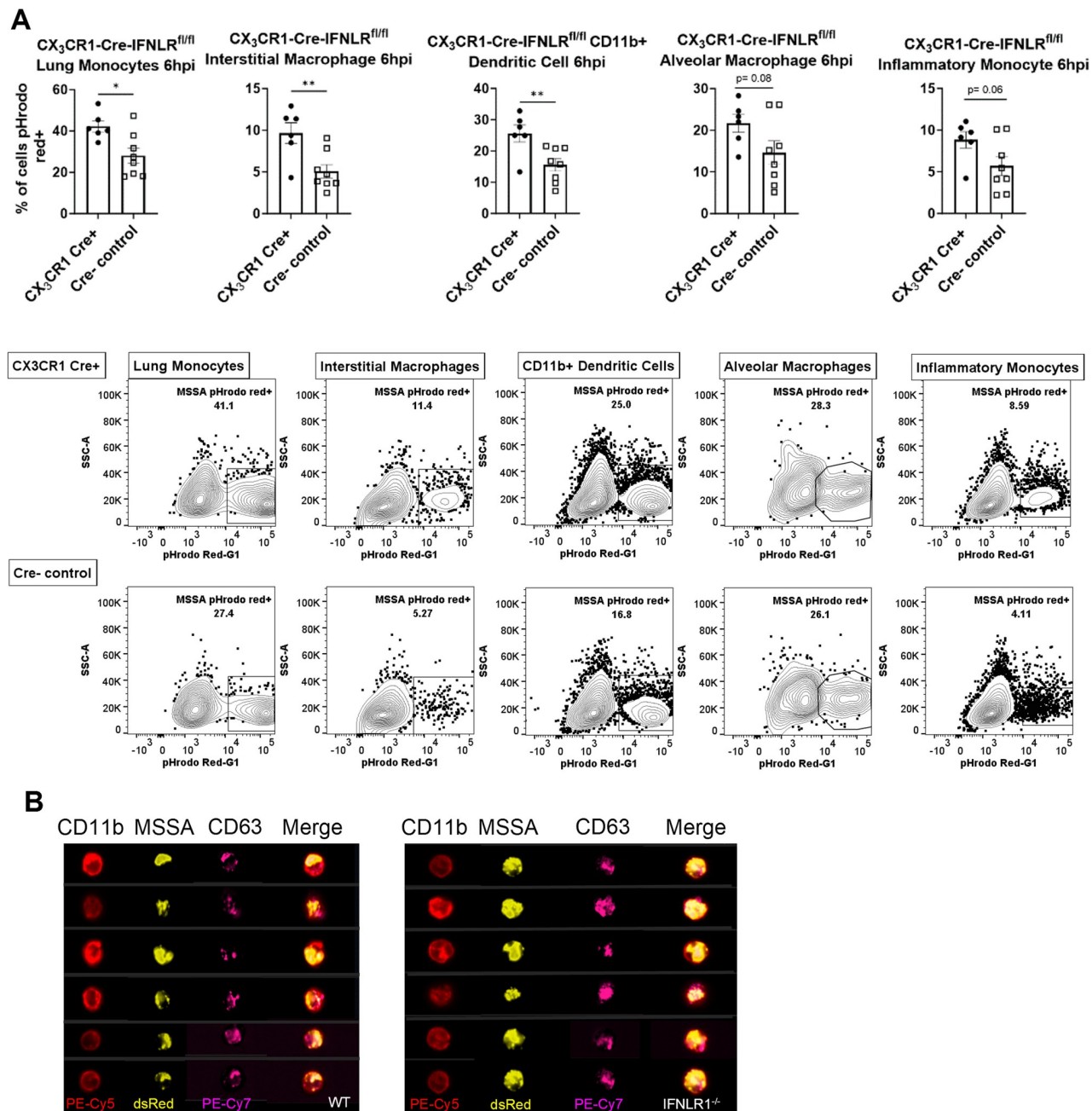

**Fig 7. Specific deletion of IFNLR1 on lung myeloid cells increases localization of bacteria within phagolysosomes.** A. CX$_3$CR1-Cre-IFNLR1$^{fl/fl}$ mice or littermate controls were super-infected as previously with pHrodo red-labeled *S. aureus* and pHrodo red+ cells were quantified by flow cytometry 6h post bacterial infection (cKO n = 7, Cre- n = 8). B. ImageStream analysis shows high colocalization of dsRed *S. aureus* and endosomal marker CD63 in global IFNLR1$^{-/-}$ and WT mice (IFNLR1$^{-/-}$ n = 8, WT n = 8). p values: *<0.05, **<0.01.

presentation (Fig 8D). These results confirm our *in vivo* data showing that IFNλ has a multi-faceted impact on the induction of antibacterial immune responses during super-infection, specifically within myeloid cells. Additionally, these data provide new avenues of study, including the potential for IFNλ to inhibit type 17 immune responses by altering antigen presentation and cell proliferation.

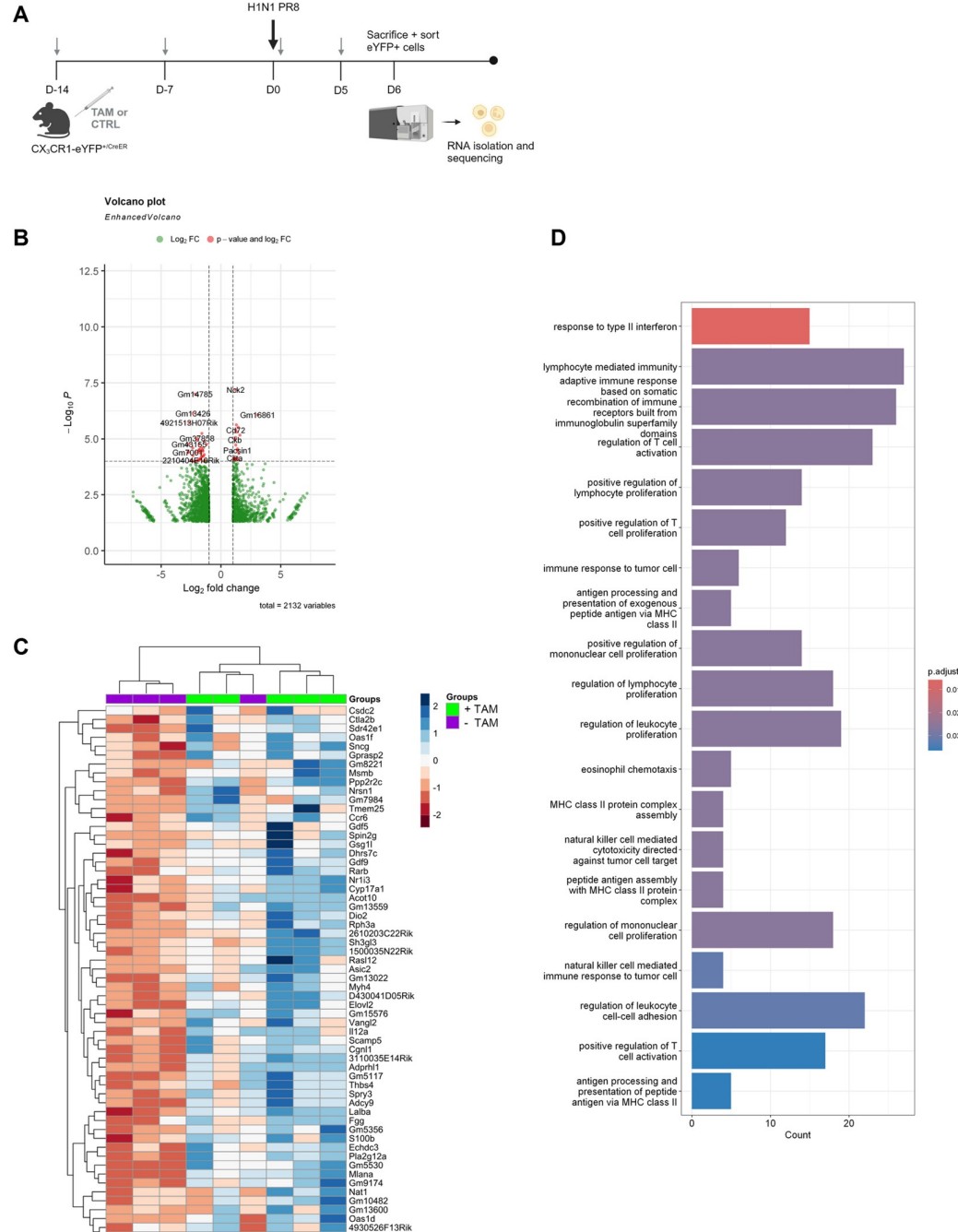

**Fig 8. Deletion of IFNLR1 in CX$_3$CR1+ immune cells impacts many immune pathways.** A. CX$_3$CR1-Cre-eYFP-IFNLR1$^{fl/fl}$ mice were treated with 2 mg Tamoxifen or corn oil controls as previously and infected with influenza for 6 days. After sacrifice, whole lung was processed into single cell suspension and sorted for eYFP+ cells before RNA isolation and bulk sequencing. In tamoxifen-treated mice, eYFP+ cells had deletion of IFNLR1 while the receptor was intact in eYFP+ cells from untreated mice (cKO: treated n = 5, untreated n = 4). Figure created on Biorender.com. B. All differentially expressed genes (DEGs) in treated mice as compared to untreated were plotted as a volcano plot. C. DEGs within the interferome were specifically analyzed, and those upregulated in treated mice were plotted as a heatmap. D. GO analysis was performed using clusterprofiler (Bioconductor) in R and significantly altered pathways in treated mice compared to controls are shown.

## Discussion

Understanding the role of antiviral immune components within the larger context of pulmonary super-infection remains important with the emergence of both seasonal and pandemic respiratory viral threats. Because IFNλ was proposed as a potential therapeutic against viruses like influenza and SARS-CoV-2 and has been reported to have divergent effects on bacterial pathogens, elucidating the interplay of IFNλ during both components of super-infection is required to ensure the safety and efficacy as a therapeutic.

In this report, we show that IFNλ inhibits bacterial clearance and antibacterial immune responses, including bacterial localization to the phagolysosome, during pulmonary super-infection. Global IFNLR1[-/-] mice have reduced bacterial burden 24 hours post bacterial infection compared to WT mice, consistent with previous reports where modulation of IFNλ alters bacterial burden [36,37,39,62]. However, IFNLR1[-/-] mice show no difference in airway protein levels or weight loss during super-infection, with only slight decreases in peribronchial inflammation in global IFNLR1[-/-] mice. As a possible explanation for enhanced clearance, we first determined levels of IL-17 and IL-22 in super-infected mice. Our laboratory has previously shown that type 17 immunity is required for clearance of *S. aureus* in the lung and that other members of the interferon family, the type I IFNs, have been shown to block the activation of type 17 immune responses during super-infection and other inflammatory disease models [33,63,64]. Additionally, a cell culture model using primary human bronchial epithelial (HBE) cells showed that IL-17 stimulation caused reductions in IFNλ1–3 production during influenza infection and in mice, treatment with IFNλ2 reduced IL-17-producing Th17 and γδ T cells in a model of collagen-induced arthritis [42,65]. Accordingly, global IFNLR1[-/-] mice had significantly increased IL-17 and IL-22 in the airways, and neutralization of IL-17A during infection caused significantly increased bacterial burden compared to untreated mice.

We theorize that enhanced IL-17 and IL-22 in our model may be due to modulation of chemokines in the lung rather than canonical cytokines upstream of IL-17 and IL-22. While IL-23, IL-1β, and IL-6 levels did not correlate with type 17 changes, we found that chemokines including CXCL16, MIP-3α, CCL17, and CCL22 were altered in super-infected global IFNLR1[-/-] mice, which could explain enhanced type 17 immune responses. However, given previous data shown in the primary HBE model, it is also possible that IL-17 and IFNλ are part of the same negative feedback loop, where the production of one would lead to inhibition of the other. A study in asthmatic mice showed that administration of inhaled IFNλ reduced levels of type 17 cytokines and Th17 population sizes by inducing IL-10-producing anti-inflammatory CD4+ T cells [66]. Global IFNLR1[-/-] mice in our study had decreased IL-10 6 hours post bacterial infection compared to WT mice, indicating that IL-10 production or lack thereof in IFNLR1[-/-] mice may also contribute to increased IL-17 levels in the lung. Our data also showed decreased IL-1β in global IFNLR1[-/-], which is consistent with other published work but also creates a disparity with the enhanced IL-17 seen in these same mice [25].

Because IFNLR1 expression is more restricted than that of type I IFN receptor, early data regarding IFNλ postulated that it may act mainly on epithelial cells at barrier sites, including the lung [41]. However, our data in mixed bone marrow chimera mice showed that IFNλ is able to directly act on immune cells as IFNLR1[-/-] immune cells showed increased phagocytosis compared to WT immune cells, resulting in normalized bacterial burden between WT and IFNLR1[-/-] hosts. These effects were more significant in WT hosts compared to IFNLR1[-/-] hosts, likely due to the importance of epithelial crosstalk during pulmonary infection. In general, the phenotypes seen in global IFNLR1[-/-] mice were more robust than phenotypes from any other model presented in this study. We hypothesize that this is due to the multifaceted and paracrine roles that IFNλ plays during infection [67]. In conditional knockout models,

IFNλ is still able to signal and induce its downstream effects in all other IFNLR1-expressing cells within the mouse. For example, neutrophils in super-infected global IFNLR1[-/-] mice showed enhanced bacterial uptake compared to WT mice, which was not recapitulated in mixed bone marrow chimeric mice or conditional S100A8-Cre-IFNLR1[fl/fl] mice. Airway epithelial cells play a vital role in protecting the lower airways and alveoli during influenza infection in a number of different ways, including via the secretion of AMPs and inflammatory cytokines like IFNλ [41,68–70]. Microarray analyses performed on alveolar type II (ATII) epithelial cells showed upregulation of many antiviral factors after influenza infection, including canonical ISGs Mx1, IFIT3, ISG15, and RSAD2, that clustered within pathways such as antigen presentation, interferon signaling, immune cell recruitment, and communication between innate and adaptive immune cells [71].

While crosstalk between epithelial cells and immune cells during infection can drive pathogen clearance, it has also been shown to inhibit resolution in a number of different cases, including pulmonary super-infection. Extracellular vesicles produced by epithelial cells during influenza infection, known to facilitate communication between cell types, were shown to increase oxidative phosphorylation and pyruvate kinase M2 expression in primary human macrophages, leading to impaired clearance of secondary *S. aureus* infection *in vitro* [72]. In other models, increased interactions between epithelial cells and immune cells correlated with more severe SARS-CoV-2 infection and impaired edema resolution after influenza infection [73,74]. Because IFNλ has been shown to impede super-infection clearance in a number of different studies, we hypothesize that the presence of IFNLR1 on other cell types, including the epithelium, in the incomplete knockout models presented dampens the phenotypes seen when compared to global IFNLR1[-/-] mice. Accordingly, S100A8-Cre-IFNLR1[fl/fl] mice did not have altered bacterial burden or survival compared to Cre- littermate controls. This was unexpected given previous reports on the interplay between IFNλ and neutrophils. Exogenous IFNλ has been shown to inhibit neutrophil recruitment and uptake of bacteria during infection, and neutrophils regulate type 17 immune responses, although the S100A8 Cre is not 100% specific to neutrophils [39,42,53,54,75]. However, these studies were performed in models where IFNλ signaling was modulated globally within the mouse, and thus communication with epithelial cells that retain their ability to respond to IFNλ could not be elucidated.

Similarly, when IFNLR1 was depleted specifically from myeloid cells using a CX$_3$CR1-Cre-IFNLR1[fl/fl] mouse line, there was not a complete phenocopy to global IFNLR1[-/-] mice. While CX$_3$CR1-Cre-IFNLR1[fl/fl] did have significant reductions in bacterial burden during super-infection compared to Cre- littermates, consistent with global IFNLR1[-/-] mice, bacterial uptake was not impacted in these conditional knockouts. Although overall uptake was not impacted, there were increased populations of pHrodo red+ cells within target myeloid populations compared to controls, indicating that phagocytes were more efficiently fusing phagolysosomes containing bacteria when IFNLR1 was knocked out. While increased phagolysosome localization would explain the decreased bacterial burden, more modest trends in other endpoints could be explained by the continued presence of IFNLR1 on other cell types within the lung. Regardless, depletion of IFNLR1 specifically on CX$_3$CR1+ cells is sufficient to protect mice against lethal super-infection. The results of this study do not elucidate individual steps of the bacterial degradation process that IFNλ may interfere with, but it is possible that IFNλ acts intrinsically on myeloid cells to impair early steps of bacterial killing and phagolysosome formation while acting extrinsically within the lung to make a sub-optimal environment for bacterial clearance.

RNA sequencing data showed that antigen presentation and MHC II assembly pathways are upregulated in tamoxifen-treated mice where IFNLR1 is depleted on CX$_3$CR1+ cells, which is another potential mechanism for regulation of type 17 immunity or bacterial

phagocytosis and degradation. These findings were particularly interesting given the limitations of this model: tamoxifen has been shown previously to impact immune responses and morbidities within mouse models and human cell culture systems [76,77]. Due to the experimental design of the bulk RNA sequencing in sorting out eYFP+ cells for analysis, sequencing a tamoxifen-treated Cre- population was not possible, as only Cre+ mice expressed the eYFP reporter. However, other data surrounding CX₃CR1-Cre-IFNLR1$^{-/-}$ mice did include tamoxifen-treated Cre- controls, which showed no modulation of the immune response specifically by tamoxifen. The side effects of tamoxifen are temporary, only occurring through the washout period, and occur primarily after doses of tamoxifen that were higher than what was given in this study, so the frequency and dosing of tamoxifen in CX₃CR1-Cre-IFNLR1$^{fl/fl}$ mice in our super-infection model is likely to minimize off-target effects. Still, further analysis into these pathways is required to more fully understand which molecules are most important for influencing antibacterial responses during super-infection.

Visualization of bacteria-positive cells by either the dsRed or pHrodo red label resulted in an interesting phenomenon in bacterial uptake. While the dot plots of pHrodo red-infected populations show a discrete positive and negative clustering of cells, flow plots of dsRed-infected cells are more continuous, with a less clear separation between dsRed+ and dsRed- cells. In order to confirm true positivity, we gated dsRed+ cells using a non-phagocytic CD90 + control population as the setting population and supported results seen in dsRed-infected global IFNLR1$^{-/-}$ mice with pHrodo red-infected mice. This difference in label visualization is likely due to the inherent kinetic differences between dsRed and pHrodo red: the RN6390 strain used in this study was transfected with a plasmid containing the dsRed gene, and therefore, is constitutively expressing dsRed. In comparison, pHrodo red fluorescence negatively correlates with the pH of the environment, remaining inactive at a high pH before phagocytosis. This difference in signal may explain discrepancies in population phenotypes, as more cells may express low levels of dsRed, either through surface binding of bacteria or early stages of phagocytosis that would not be reflected in pHrodo red+ populations. ImageStream analyses of cells from dsRed-RN6390-infected mice were critical in distinguishing between internalized bacteria and extraneous signals, and while these data were analyzed using solely internalized dsRed signal, it is possible that the total number of cells expressing any level of dsRed was higher than what was distinguishable in mice infected with pHrodo red-labeled bacteria.

Overall, identifying components of the antibacterial immune response that are altered by IFNλ can inform clinicians on how best to use IFNλ as an antiviral treatment in humans. While we have shown here that IFNλ signaling impacts phagocyte function and inhibits IL-17-mediated immunity to prolong bacterial clearance during super-infection, additional work is required to further parse out impacts of IFNλ that may be cell- or timing-specific. By fully understanding the inhibitory functions of IFNλ, its therapeutic potential can be modulated and optimized against respiratory viruses.

## Materials and methods

### Ethics statement

All experiments were conducted with approval from the University of Pittsburgh Institutional Animal Care and Use Committee (Protocol #23073501).

### Mice

IFNLR1$^{-/-}$ and IFNLR1$^{fl/fl}$ mice were generated by the Innovative Technologies Development Core at the University of Pittsburgh and maintained under pathogen free conditions at UPMC Children's Hospital of Pittsburgh. Wild type (WT) C57BL/6 mice (Taconic Farms,

Germantown, NY) were purchased as age- and sex-matched controls to global IFNLR1$^{-/-}$ mice. On day 0, six to eight-week old mice were infected with 900 PFU of mouse-adapted influenza A/PR/8/34 H1N1, generously provided by Dr. Radha Gopal at the University of Pittsburgh. Six days later, mice were challenged with 5 x 10$^7$ CFU USA300 MRSA or RN6390 MSSA containing a fluorescent chloramphenicol-resistant dsRed plasmid, gifted by Dr. Robert Shanks at the University of Pittsburgh. For survival experiments, mice were given a lethal bacteria dose of 2.5 x 10$^8$ CFU USA300 MRSA and monitored twice daily for weight loss and visible signs of illness (hunched posture, ruffled fur, temperature decrease, low movement, etc.). Mice were euthanized after weight loss greater than 30% or when visibly distressed. All infections were given after isoflurane anesthetization by oropharyngeal aspiration and mice were euthanized via pentobarbital injection 6 or 24 hours after bacterial infection. CX$_3$CR1$^{CreER}$ and MRP8 (S100A8)-Cre-ires mice were purchased from The Jackson Laboratory (Bar Harbor, ME) and crossed with IFNLR1$^{fl/fl}$ mice to generate homozygous floxed, Cre+ heterozygous experimental mice. CX$_3$CR1-Cre-IFNLR1$^{fl/fl}$ and littermate controls were treated intraperitoneally with 2 mg Tamoxifen dissolved in 100% ethanol and diluted in corn oil. Mice were treated 14 days and 7 days before influenza infection, on the day of influenza infection, and on day 5 of super-infection model. In experiments where IL-17A was neutralized, mice were treated with 200 μg anti-IL-17A (17F3) or mouse IgG1 isotype controls (MOPC-21; BioXCell, Lebanon, NH) on days 1, 3, and 5 post-influenza infection.

## Bacterial plating

The right upper lobe was collected from mice and homogenized in 1mL of sterile PBS. After, lung homogenate was serial diluted and plated in 10μL dots on culture plates and incubated overnight at 37˚ Celsius. Bacterial CFU/mL was identified by colony counting.

## Bronchoalveolar lavage fluid collection

Bronchoalveolar lavage (BAL) fluid was collected and processed as described previously [78]. Briefly, after lethal injection with pentobarbital, mice were cannulated and lavaged with 1 mL of PBS. BAL fluid was spun down, supernatant collected for downstream analyses, and cells treated with ACK lysis buffer (Gibco Fisher Science, Hampton, NH) to lyse red blood cells. Cell pellets were resuspended and counted using a hemocytometer.

## Flow cytometry

Lung lobes were collected from mice and dissected using sterile instruments before digestion in 1 mg/mL collagenase media for 1 hour at 37˚ Celsius (DMEM, Gibco Fisher Scientific Hampton, NH). After digestion, lungs were filtered into single cell suspension using a 70 μm filter and treated with ACK lysis buffer (Gibco Fisher Scientific, Hampton, NH) to remove red blood cells. Cells were then resuspended in PBS and stained as follows for conventional flow cytometry analysis using the BD Fortessa. Cells were stained with anti-CD45 (30-F11, BD Pharmingen, San Diego, CA), CD90 (53–2.1, BD Biosciences, Franklin Lakes, NJ), B220 (RA3-6B2, BD Pharmingen, San Diego, CA), CX$_3$CR1 (SA011F11, BioLegend, San Diego, CA), CD11b (M1/70, BioLegend, San Diego, CA), Siglec F (E50-2440, BD Biosciences, Franklin Lakes, NJ), MHC-II (M5/114.15.2, BD Biosciences, Franklin Lakes, NJ), CD11c (HL3, BD Biosciences, Franklin Lakes, NJ), NK1.1 (PK136, BioLegend, San Diego, CA), CD103 (M290, BD Biosciences, Franklin Lakes, NJ), Ly6G (1A8, BD Pharmingen, San Diego, CA), Ly6C (HK1.4, BioLegend, San Diego, CA), and F4/80 (T45-2342, BD Biosciences, Franklin Lakes, NJ). The viability dye Ghost Dye Red 780 (Cytek Biosciences, Fremont, CA) was used to distinguish live and dead cells, and staining buffers included Super Bright Complete Staining Buffer

(ThermoFisher Scientific, Waltham, MA), and True-Stain Monocyte Blocker (BioLegend, San Diego, CA). Absolute cell counts were determined using CountBright Plus Absolute Counting Beads (ThermoFisher Scientific, Waltham, MA) and analysis was performed on FlowJo. Gating for specific myeloid populations was performed as outlined in S9 Fig.

For ImageStream analyses, samples were processed as above and stained with CD45 (30-F11, Invitrogen-ThermoFisher Scientific, Waltham, MA), CD11b (M1/70, BioLegend, San Diego, CA), CD63 (NVG-2, Biolegend, San Diego, CA), $CX_3CR1$ (SA011F11, BioLegend, San Diego, CA), and viability dye Ghost Dye Red 780 before analysis on the Amnis ImageStream.

## Histology

Left lung lobes were collected from mice and inflated with and preserved in a 10% formalin solution. Fixed tissues were transferred to 70% ethanol and sent to the Histology Core at the University of Pittsburgh to be paraffin embedded, sectioned, and stained for analysis using hematoxylin and eosin staining (H&E). Slides were then scanned using the Leica Aperio CX2 digital slide scanner and using QuPath analysis software. Pathologic scoring was based on perivascular, peribronchial, and parenchymal damage and plotted as a ratio of inflamed to healthy tissue area.

## Mixed bone marrow chimeras

WT and IFNLR1$^{-/-}$ mice were sub-lethally irradiated with two doses at 5 Gy with 4 hours in between dosing. During irradiation, bone marrow cells were collected from WT and IFNLR1$^{-/-}$ mice. Briefly, mice were euthanized by carbon dioxide and cervical dislocation; femurs were collected and flushed with media (RPMI, Gibco Fisher Scientific, Hampton, NH) to collect bone marrow. Cells were pelleted by centrifugation at 420g for 7 minutes, treated with ACK lysis buffer, and filtered through a sterile 70 μm filter. Irradiated mice were then intravenously injected with a 1:1 ratio of WT and IFNLR1$^{-/-}$ cells to total $1 \times 10^7$ cells and allowed to reconstitute for 9 weeks, before infection as described above.

## Lincoplex and protein assays

Cytokine production was measured and quantified in lung homogenate of infected mice by using the Bio-Plex Mouse Chemokine Panel 31-plex kit (Bio-Rad, Hercules, CA) run on the Luminex Magpix multiplexing platform. Lung leak was assessed by measuring protein concentration in BAL fluid using the Pierce BCA protein assay kit (ThermoFisher Scientific, Waltham, MA) as directed. ELISA DuoSet kits for mouse IL-17 and IL-22 were purchased from R&D Systems (Biotechne, Minneapolis, MN) and performed to the manufacturer instructions.

## RNA isolation and bulk RNA sequencing

Whole lung lobes were snap frozen from $CX_3CR1$-Cre-IFNLR1$^{fl/fl}$ mice treated with Tamoxifen or corn oil controls and homogenized in RLT Buffer with β-mercaptoethanol. RNA was isolated using the Qiagen RNeasy Mini Kit as instructed (Qiagen, Hilden, Germany). cDNA was synthesized as directed using the iScript cDNA synthesis kit (Bio-Rad, Hercules, CA). qPCR was run using SsoAdvanced universal probes supermix (Bio-Rad, Hercules, CA), and probes used were TaqMan real-time PCR assay primer probes that were target specific (ThermoFisher Scientific, Waltham, MA). Viral burden was quantified by qRT-PCR for viral matrix protein on whole lung RNA as described elsewhere [79,80]. Forward Primer:5′-GGACTGCA GCGTAGACGCTT-3′, Reverse Primer:5′- CATCCTGTTGTATATGAGGCCCAT-3′, Probe:5′-/56-FAM/CTCAGTTAT/ZEN/TCTGCTGGTGCACTTGCCA/3IABkFQ/−3′. Data

shown as fold changed was normalized to both a housekeeping gene and to control samples. Sequencing was performed and analyzed by MedGenome, Inc (Foster City, CA) using the Takara SMART-Seq v4 Ultra Low Input RNA kit (Takara Bio, Kusatsu, Japan): 2x75 bp, 20 million paired ends reads (40M total). Additional analyses were performed using clusterprofiler package in R (Bioconductor) to visualize differentially expressed genes between treated and untreated mice. All sequencing data has been uploaded to Gene Expression Omnibus (GSE268994).

## Statistical analysis

Data were analyzed using GraphPad Prism (San Diego, CA), and experiments were repeated a minimum of 3 times unless indicated. All data are presented as mean with SEM unless otherwise noted, and statistical testing included unpaired t-testing or one-way ANOVA with multiple comparisons as appropriate. Statistical significance was defined as having a p value less than or equal to 0.05.

## Supporting information

**S1 Fig. Global IFNLR1-/- mice are comparable to WT at early super-infection timepoints.** A. Super-infected global IFNLR1-/- mice show trending reductions in bacterial burden 6 hours post bacterial infection compared to WT mice (IFNLR1-/- n = 6, WT n = 6, data from 2 replicates). B. Protein levels in the airways of super-infected mice were evaluated 6 hours post super-infection onset (IFNLR1-/- n = 10, WT n = 10). C. Mouse weight loss was monitored daily during infection time-course (data from 2 replicates). D. Immune cell infiltration into the airways was assessed by quantification of cells in the BAL fluid 6 hours post super-infection (IFNLR1-/- n = 4, WT n = 4, representative data from 2 replicates). E. Global cytokine levels were detected using Bio-Plex assays in super-infected mice 6 hours post bacterial infection (IFNLR1-/- n = 10, WT n = 10). p values: *<0.05, **<0.01, ***<0.001, ****<0.0001.
(PDF)

**S2 Fig. Global IFNLR1-/- alters immunopathology after super-infection.** A. Pathology scores were determined by Qupath software analysis of perivascular, peribronchial, and parenchymal lung damage (IFNLR1-/- n = 11, WT n = 12). B. Representative images of whole lung slides after H&E staining (IFNLR1-/- n = 2, WT n = 2). C. Levels of *ifnlr1* transcript were assessed in WT and IFNLR1-/- CD11c+ cells by qPCR. Cells were sorted from naïve mice (n = 4). p values: *<0.05, **<0.01, ***<0.001, ****<0.0001.
(PDF)

**S3 Fig. IFNλ disrupts phagocytosis of multiple phagocyte populations during super-infection.** A. Representative flow plots of non-phagocytic CD90+ cells show gating strategy for dsRed+ cells in WT (left) and IFNLR1-/- mice (right). B. Frequency (left) and total cell counts (right) of dsRed+ CD11b+ dendritic cells (DCs; CD45+, CD11c+, CD11b+, CD103-) are increased in IFNLR1-/- mice compared to WT during super-infection (representative data, n = 4 for all groups). C. Total cell counts of dsRed+ inflammatory monocytes, lung monocytes, interstitial macrophages, and neutrophils largely recapitulate frequencies of dsRed+ cells seen in Fig 2A (representative data, n = 4 for all groups). D. Overall immune cell recruitment measured by flow cytometry showed no differences in cell count (top) or frequency (bottom) in super-infected IFNLR1-/- versus WT mice (top: IFNLR1-/- n = 8, WT n = 8; bottom: IFNLR1-/- n = 14, WT n = 14). p values: *<0.05, **<0.01, ***<0.001, ****<0.0001.
(PDF)

**S4 Fig. IFNλ disrupts phagocytosis of multiple phagocyte populations during super-infection.** A. Frequencies of pHrodo red+ cells in global IFNLR1-/- and WT mice during super-infection (n = 6). B. Representative flow plots from super-infected global IFNLR1-/- (top) and WT (bottom) mice. Mice were infected with 900 PFU PR8 and 5 x 107 CFU pHrodo-red labeled MRSA strain USA300. Global IFNLR1-/- mice have increased uptake of bacteria as quantified by pHrodo red+ cells compared to WT mice, consistent with Fig 3A. (PDF)

**S5 Fig. Mixed bone marrow chimeric reconstitution of WT and IFNLR1-/- hosts.** A. WT or IFNLR1-/- mice were sublethally irradiated and reconstituted with 1:1 WT:IFNLR1-/- bone marrow for 9 weeks before being super-infected. Figure created on Biorender.com. B. Reconstitution of bone marrow cells was comparable between WT and IFNLR1-/- cells in both mouse backgrounds. C. Total counts of immune cell recruitment to the lung was not significantly altered by host genotype or immune cell genotype, with the exception of neutrophils in IFNLR1-/- hosts (host: KO n = 8, WT n = 8). p values: $^*$<0.05, $^{**}$<0.01, $^{***}$<0.001, $^{****}$<0.0001. (PDF)

**S6 Fig. Influenza controls show no difference in S100A8-Cre-IFNLR1fl/fl mice compared to controls.** A. S100A8-Cre-IFNLR1f l/f l mice show no difference in influenza *m* transcript levels when infected with influenza alone compared to Cre- controls (n = 6). B. Weight loss between S100A8-Cre- IFNLR1f l/f l and Cre- controls is unchanged during single influenza infection (n = 6). C. Weight loss between S100A8-Cre-IFNLR1f l/f l and Cre- controls is unchanged during single 24 hour MRSA infection (cKO n = 7, Cre- n = 6). D. Production of type I IFNs (*ifnb*) is not altered between S100A8-Cre-IFNLR1f l/f l mice and Cre- controls during super-infection, influenza, or MRSA infection. (PDF)

**S7 Fig. CX3CR1-specific depletion of IFNLR1 does not largely impact endpoints after super-infection, lowers viral burden.** A. CX3CR1 Cre specificity was tested in Cre+ mice (cKO) after Tamoxifen injections through detection of an eYFP reporter by flow cytometry (cKO n = 5, Cre- n = 6). B. *ifnlr1* transcripts were measured by qPCR in tamoxifen-treated and–untreated mice (treated n = 4, untreated n = 4) C. Influenza *m* transcript levels were shown to be significantly reduced in CX3CR1-Cre-IFNLR1f l/f l mice compared to Cre-controls during single influenza infection. D. CX3CR1-Cre-IFNLR1f l/f l mice or Cre- controls were infected with influenza alone and show comparable weight loss over infection timecourse (n = 5). E. Production of type I IFNs (*ifnb*) is not altered between CX3CR1-Cre-IFNLR1f l/f l mice and Cre- controls during single influenza infection (n = 5). F. Transcripts of *sftpc*, *scgb1a1*, and *tjp1* were measured in lungs of cKO and control mice 24 hours after *S. aureus* infection (cKO n = 8, Cre- n = 8). All data is from 2 replicates. p values: $^*$<0.05, $^{**}$<0.01, $^{***}$<0.001, $^{****}$<0.0001. (PDF)

**S8 Fig. CX3CR1-specific depletion of IFNLR1 does not impact bacterial uptake.** A. Frequencies of dsRed+ phagocytes was not altered between cKO mice and Cre- controls at 6h (open bars) or 12h (hashed bars) after bacterial infection (6h open bars: cKO n = 7, Cre- n = 7; 12h hashed bars: cKO n = 6, Cre- n = 5). B. Bacterial burden at earlier timepoints of 6h (left) or 12h (right) show no difference between CX3CR1-Cre-IFNLR1fl/fl mice and Cre- controls (6h open bars: cKO n = 7, Cre- n = 7; 12h hashed bars: cKO n = 6, Cre- n = 5). All data is from two replicates. p values: $^*$<0.05, $^{**}$<0.01, $^{***}$<0.001, $^{****}$<0.0001. (PDF)

**S9 Fig. Gating strategy for myeloid cell subsets.** Flow cytometry gating for immune cell populations was performed as outlined.
(PDF)

**S1 Data.** Raw data files for main manuscript figures. Xcel file containing the raw data for Figures 1–8.
(XLSX)

**S2 Data.** Raw data files for supplemental figures. Xcel file containing the raw data for Supplemental Figures 1–9.
(XLSX)

## Acknowledgments

The authors wish to thank the UPMC Children's Hospital of Pittsburgh Flow Cytometry Core and the University of Pittsburgh Unified Flow Cytometry Core for expertise and technical assistance with the studies herein. We would also like to thank Dr. Sebastian Gingras, the Director of the University of Pittsburgh Innovative Technologies Development Core Facility. Dr. Gingras was essential to generation of the IFNLR1 floxed mice. Lastly, we thank Neil Patel and Marco Corbo from MedGenome for their help and support with the RNA sequencing project.

## Author Contributions

**Conceptualization:** Danielle Antos, John F. Alcorn.

**Data curation:** Danielle Antos.

**Formal analysis:** Danielle Antos, John F. Alcorn.

**Funding acquisition:** Danielle Antos, John F. Alcorn.

**Investigation:** Danielle Antos, Olivia B. Parks, Alexis M. Duray, Nevil Abraham, Joshua J. Michel, Saran Kupul, Rosemary Westcott.

**Methodology:** Danielle Antos, Olivia B. Parks, Nevil Abraham, Joshua J. Michel, Saran Kupul, Rosemary Westcott.

**Project administration:** John F. Alcorn.

**Resources:** John F. Alcorn.

**Supervision:** John F. Alcorn.

**Validation:** Danielle Antos.

**Visualization:** Danielle Antos.

**Writing – original draft:** Danielle Antos, John F. Alcorn.

**Writing – review & editing:** Danielle Antos, John F. Alcorn.

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
