## [Decision Letter · Decision Letter 0]

2 Apr 2024

Dear Dr. Alcorn,

Thank you very much for submitting your manuscript "Cell-intrinsic regulation of phagocyte function by interferon lambda during pulmonary viral, bacterial super-infection" for consideration at PLOS Pathogens. As with all papers reviewed by the journal, your manuscript was reviewed by members of the editorial board and by several independent reviewers. In light of the reviews (below this email), we would like to invite the resubmission of a significantly-revised version that takes into account the reviewers' comments.

We cannot make any decision about publication until we have seen the revised manuscript and your response to the reviewers' comments. Your revised manuscript is also likely to be sent to reviewers for further evaluation.

Sincerely,

Jacob S. Yount

Academic Editor

PLOS Pathogens

Kanta Subbarao

Section Editor

PLOS Pathogens

Michael Malim

Editor-in-Chief

PLOS Pathogens

orcid.org/0000-0002-7699-2064

Reviewer's Responses to Questions

**Part I - Summary**

Reviewer #1: Antos, et. al. investigates a role for IFN-lambda in regulation of S. aureus bacterial superinfection following influenza A virus (IAV) infection. The authors find IFN-lambda promotes bacterial burden following superinfection. They identify enhanced IL-17 production in mice lacking Ifnlr1, suggesting low IL-17 level leads to impairment of bacterial uptake in the presence IFN-lambda signaling. They claim there is no intrinsic role for IFN-lambda in neutrophil bacterial uptake, but it is instead Cx3CR1-expressing cells that have enhanced uptake in the absence of IFN-lambda signaling to control bacterial superinfection. Overall, this manuscript presents an interesting finding expanding the fields understanding of the role of IFN-lambda in immune cells to modulate pulmonary infection outcomes. However, this manuscript lacks critical controls, experimental details, and discussion of the existing literature in the context of this work to support the conclusions made by the authors.

Reviewer #2: The authors show that IFNLR1 KO mice are protected in the context of secondary MRSA superinfection after influenza infection, and have enhanced bacterial clearance. They identify IFN-λ activity on CX3CR1 phagocytes as responsible for the enhanced bacterial clearance. Mechanistically they claim that CX3XR1 cells devoid of IFNLR1, phagocytose bacteria more efficiently and thus enhance global bacterial clearance. The authors also exclude a role for neutrophils and epithelial cells with genetic expeirments.

While the concept put forward by the authors, of an immune regulatory role of IFN-λ independent of its action on neutrophils, is intriguing, the study lacks rigor, appropriate controls and presents several problems, especially since its claims are contrary to the current literature and, therefore, need particular attention and rigor to exclude any artifact or overinterpretation of data.

Particularly puzzling are:

the low magnitude of the observed effects, namely in terms of morbidity/survival;

the large variability between experiments regarding the principal readout, i.e. MRSA phagocytosis, that varies from 87% of phagocytosing cells in one experiment to 7% in the next ;

the absence of rigorous controls, e.g. the authors never show a facs plot of immune cells from uninfected mice.

Importantly, the authors never show that CX3CR1+ cells express IFN-λ receptor and that they respond to it. They merely show non-significant trends in the expression of IFNLR1 in “tamoxifen treated and untreated mice” (no indication on the tissue/cell type analyzed). Moreover, when they perform bulk RNAseq they fiund a limited number of differentially expressed genes, none of them is a bona fide ISG, and the volcano plot shows that genes that are significantly different (above the pValue threshold) have a very low fold change (<2,5). These data are not convincing me of the fact that CX3CR1 phagocyte respond to IFN-λ.

Please find below the major specific points .

These are significant shortcoming. This paper in the present form is unsuited for publication.

Reviewer #3: This paper deals with the interesting question of whether IFN lambda (IFNL) can have direct effects on non-epithelial cells. As IFNL receptor (IFNLR) expression is still controversial on immune cells except for mouse neutrophils, this is a timely report trying to address this in the context of influenza – Staph superinfection. The study is an honest account of data that partly converge towards the same direction but also are contradictory in many aspects, so that forming a cohesive picture of what happens in the absence of IFNLR signalling is impossible. Overall, every single major concern below needs to be addressed thoroughly to render this study robust enough for publication.

**Part II – Major Issues: Key Experiments Required for Acceptance**

Reviewer #1: For all experiments mice are all infected with 900 PFU PR8, but what this represents in terms of lethality on its own is not included in the manuscript. For example, in Figure 1, IAV only controls are still losing weight at the conclusion of the experiment and it is unclear whether superinfection impacts mortality. IAV-only controls are also critical for interpretation of Figures 5 and 6 where the cell types targeted are known to be important for IAV infection on its own.

The S100A8-Cre mice are not included in the methods section. However, it seems, according to Jax, that these are the common name for Mrp8-Cre mice known not to be neutrophil specific. This critical point should be clarified.

In addition, the disconnect between enhanced MSSA uptake in neutrophils in the global KO and no difference in the outcome of Figure 5 should be discussed in greater detail

In general, the manuscript is under-cited. There are many publications that have investigated the role of IFN-lambda in regulating IAV pathogenesis and describing the role of IFN-lambda in immune cell functions (even in the context of IAV infection) that are not included or discussed. Importantly, how these defects during IAV-only potentially contribute to the defects observed during bacterial superinfection are not discussed.

Reviewer #2: Major

I do not understand what the authors claim in figure 2. Slight differences in IL17 and Il22 levels are detected, Il-17 delivery has similar effects in WT and IFNLR1 ko (in an experiment in which WT and KO do not differ in bacterial burden) and blockade of CXCL16 (unclear why it was chosen) doesn’t affect bacterial burden. The involvement of the communication between IFN-λ and IL-17 pathway in this process is unclear.

In figure 3 ( and throughout) the important control of uninfected mice is missing, especially for the FACS plots. How did thea author chose where to place the threshold for phagocytosis vs no phagocytosis? Additionally, it doesn’t seem like a population is picking up dsRED MRSA while another is not, but rather that there’s a global shift in the whole sample. Is it because everyonbe is phagocytosing a bacterium with low fluorescence? Are there any cell types (i.e. T cells) that don’t phagocytose to show as control? Most puzzling is that while the authors here claim that upto 90% of the cells phagocytose mrsa at 24h, in the very next figure, the authors show a phagocytosis rate of 7% at best. Imagestream analyses are important in this context but why did the author not show a FACS-like dot plot from this experiment? And why did they only show 4 positive cells?

In figure 4 Along with the problem of the vastly different % between experiments the differences between WT and KO cells are modest an not always statistically significant.

In figure 6, as the most important readout is phagocytosis, why didn’t they show facs plots? Where are the statistical analyses of fig. 6E? What is the significance of the IDO expression? Is the effect dependent on phagocytosis or epithelial repair?

Finally in figure 7 the authors show a plot in which phagocytosing cells and non-phagocytosing cells are clearly separated. Why didn’t the author use the same model throughout? (pHrodo labeled bacteria?). While the data seem more solid ( despite only trends in certain populations) it is unclear whether the analyzed cells respond to IFN-λ at all. In fact, the RNAseq data don’t seem to support a cell intrinsic role for IFN-λ. For starters, among the differentially expressed genes, IFNLR1 doesn’t figure. The genes differentially expressed and above the pval threshold are very few, their fold change is very low and I could not identify canonical ISGs. Moreover, the genes in the heatmap are not the same than the volcano plot, how were they identified?

Reviewer #3: Major concerns:

1. Bacterial infection model: The authors state that they infect with 5 x 10e7 CFUs of Staph, of which only a fraction is recovered at 6 h (assuming a volume of roughly 1 ml of homogenate), with a further roughly 100 fold reduction between 6 and 24 h (Fig. 1B). Do these bacteria proliferate at all in the mice or would applying heat-inactivated bacteria give the same picture? While the difficulties with bacterial infection models are well appreciated, the numbers here do not suggest that Staph infects these mice particularly well, limiting the robustness of the infection model. In 1C, it appears as if bacterial superinfection protects the wt mice from barrier damage vs single viral infection – how can this be explained? None of the known superinfection models seem to show such a protective effect. Also, wt mice are worse off than IFNLR KO in single viral infection – how come? While redundancy between IFN systems suggest that there may be no dramatic difference between wt and IFNLR KOs, this result is not in line with published data. For data in Fig. 1E, F it is not indicated when these measurements were done, and more generally, to make statements about inflammation, time course analysis of these data is needed. In addition, the bacterial load differences that are the key difference between wt and KO mice as stated by the authors already disappear in the control groups in Fig. 2B: here, the empty vector controls show no significant difference between wt and KO.

Overall, the base line infection data here leave important doubts about the model, its robustness and its predictability.

2. Link to type 17 responses:

Fig. 2A, when is this measured? The authors need to keep in mind that bacterial burden drops very rapidly and that cytokine changes may be very dynamic, so this is difficult to determine by looking at just one time point. As pointed out already above, in Fig. 2B, the base line difference between wt and KO should be preserved in the GFP vector – treated groups but this is not the case. Furthermore, the only significant difference here is in the KOs +/- additional IL-17 - but these are the mice that already have more IL-17, while wt mice with lower IL-17 levels to start with should have a stronger effect from IL-17 addition, which is not the case – this set of data does not support much the authors’ argument. In addition, compared to the massive differences in CFUs found in other superinfection settings, the differences here are subtle , and it is difficult to understand how much biological significance there is in these small differences.

Fig. 2D,E: CXCL16 should BLOCK IL17/22 to prove the authors’ point, however the authors show only CFU data, not IL-17 protein, and il22 only by RNA. All this data is required to support the points being made here. MIP3a is not modifiec experimentally, why?

3. Link to phagocytosis:

In fig.3, how are cell subsets identified / pre-gated? The data shown here indicates that there is an overall shift in fluorescence, not two populations – how come? This looks like a 100:0 result which does not make much sense. In addition, the BM chimeras in fig.4 show Staph labelling that is very low – none of the high percentages of the full KO are seen here – how is this possible? The pHrodo data gives two clear-cut populations, this should be equally the case with the plain labelled bacteria. All this does not induce much confidence in this reviewer that the bacterial loading data is reproducible or interpretable.

Within the experiment shown in fig.4, the differences may be statistically significant, but they are very very small - how do we know they actually have biological meaning? In addition, it is unclear to this referee why would this (small) difference in BM-derived cells would disappear in the KO recipient background. Overall these data are inconclusive as to the role of phagocytosis downstream on IFNL.

4. This lack of conclusiveness is further compounded when differences in phagocytic capacities are not reproduces in myeloid-specific IFNL KO mice (Fig.6E): At this point, everything that the authors suggest up to here essentially collapses, and other, unrelated explanations are put forward, such as increased IDO production in wt vs. KO mice: this comes a bit out of the blue here and is completely unlinked to all prior data. In contrast, the increased bacterial burden (Fig. 6B) in CX3CR1 Cre KOs phenocopies full KOs which is fine, but why then do the mixed BM chimeras not show the same? Overall, three different models (full KO, BM chimeras with KO BM, myeloid cell-specific conditional KO) give three different outcomes, and the overlap is too partial to draw a conclusive picture of what is happening here.

5. Have the authors checked type I IFN levels in the IFNRL KOs, reduced type I IFN could explain most findings here…

Have the authors checked IFNLR levels on all the cells you are talking about here?

6. In fig. 7, authors should do these experiment with BALF-derived MPh, no need to use an artificial model such as THP-1 cells. If they want to include human data they should use blood monocytes and MPh derived from them.

7. Fig.8: Tamoxifen has clear effects on immune cells, so another key control that needs to be included is +TAM in Cre-neg cells. Fig.8A: why no unsupervised clustering for treatment samples?

**Part III – Minor Issues: Editorial and Data Presentation Modifications**

Reviewer #1: It would be useful to determine viral copy number prior to superinfection on day 6 to determine the defect in IAV-only controls.

Why IL-17 is investigated when there are no significant differences observed in Figure 1 is not discussed.

Reviewer #2: Minor

Fig1F, I have no info to interpret the statistics. What was compared?

Thp1 cell experiments are not convincing at all, differences between 0.9 and 1 % if phagocytosing cells, despite having statistical significance, are not biologically relevant (see Fig 7C). do Thp1 cells even respond to IFN-λ? Did they measure?

Overall the experiments, sample size and statistics are not described in a sufficiently clear manner.

Reviewer #3: Minor concerns:

Line 106 fix this phrase

Line 130 rephrase, IFNL does NOT mediate bact clearance

Line 332 no you do not show this – it does not happen in the myeloid- or NPh-specific IFNLR KOs.

Fig. 6C: p=0.2 is not an indication of significant differences, this would happen by chance in one of 5 attempts…

PLOS authors have the option to publish the peer review history of their article (what does this mean?). If published, this will include your full peer review and any attached files.

Reviewer #1: No

Reviewer #2: No

Reviewer #3: No
---

## [Decision Letter · Decision Letter 1]

1 Aug 2024

Dear Dr. Alcorn,

Thank you very much for submitting your manuscript "Cell-intrinsic regulation of phagocyte function by interferon lambda during pulmonary viral, bacterial super-infection" for consideration at PLOS Pathogens. As with all papers reviewed by the journal, your manuscript was reviewed by members of the editorial board and by several independent reviewers. The reviewers appreciated the attention to an important topic. Based on the reviews, we are likely to accept this manuscript for publication, providing that you modify the manuscript according to the review recommendations.

Sincerely,

Jacob S. Yount

Academic Editor

PLOS Pathogens

Kanta Subbarao

Section Editor

PLOS Pathogens

Michael Malim

Editor-in-Chief

PLOS Pathogens

orcid.org/0000-0002-7699-2064

Reviewer Comments (if any, and for reference):

Reviewer's Responses to Questions

**Part I - Summary**

Reviewer #1: This manuscript has been significantly improved and the authors have sufficiently addressed all reviewer comments. GEO Accession is still not available for RNAseq data set and deposition in repository should be completed prior to/at the time of publication.

**Part II – Major Issues: Key Experiments Required for Acceptance**

Reviewer #1: (No Response)

**Part III – Minor Issues: Editorial and Data Presentation Modifications**

Reviewer #1: (No Response)

PLOS authors have the option to publish the peer review history of their article (what does this mean?). If published, this will include your full peer review and any attached files.

Reviewer #1: No

Figure Files:

Data Requirements:

Reproducibility:

References:

---

## [Editor Report · Decision Letter 2]

12 Aug 2024

Dear Dr. Alcorn,

We are pleased to inform you that your manuscript 'Cell-intrinsic regulation of phagocyte function by interferon lambda during pulmonary viral, bacterial super-infection' has been provisionally accepted for publication in PLOS Pathogens.

Best regards,

Jacob S. Yount

Academic Editor

PLOS Pathogens

Kanta Subbarao

Section Editor

PLOS Pathogens

Michael Malim

Editor-in-Chief

PLOS Pathogens

orcid.org/0000-0002-7699-2064
---

## [Editor Report · Acceptance letter]

20 Aug 2024

Dear Dr. Alcorn,

We are delighted to inform you that your manuscript, "Cell-intrinsic regulation of phagocyte function by interferon lambda during pulmonary viral, bacterial super-infection," has been formally accepted for publication in PLOS Pathogens.

Best regards,

Michael Malim

Editor-in-Chief

PLOS Pathogens

orcid.org/0000-0002-7699-2064